# Interpretability Needs a New Paradigm

## Abstract

Interpretability is the study of explaining models in understandable terms to humans. At present, interpretability is divided into two paradigms: the intrinsic paradigm, which believes that only models designed to be explained can be explained, and the post-hoc paradigm, which believes that black-box models can be explained. At the core of this debate is how each paradigm ensures its explanations are *faithful*, i.e., true to the model's behavior. This is important, as false but convincing explanations lead to unsupported confidence in artificial intelligence (AI), which can be dangerous. This paper's position is that we should think about alternative paradigms while staying vigilant regarding faithfulness. First, by examining the history of paradigms in science, we see that paradigms are constantly evolving. Then, by examining the current paradigms, we can understand their underlying beliefs, the value they bring, and their limitations. Finally, this paper presents 3 emerging paradigms for interpretability. The first paradigm designs models such that faithfulness can be easily measured. Another optimizes models such that explanations become faithful. The last paradigm proposes to develop models that produce both a prediction and an explanation.

Rev.3

## 1 Introduction

In 1874, Georg Cantor proposed set theory and showed there exists at least two kinds of infinity. This divided the mathematical field. The Intuitionists, who named Cantor's theory nonsense, thought that math was a pure creation of the mind and that these infinities weren't real. Henri Poincaré said: "Later generations will regard Mengenlehre (set theory) as a disease from which one has recovered" (Gray, 1991). Leopold Kronecker called Cantor a "scientific charlatan" and "corruptor of the youth" (Dauben, 1977).

The other group, the Formalists, thought that by using Cantor's set theory, all math could be proven from this fundamental foundation. David Hilbert said: "No one shall expel us from the paradise that Candor has created" (Hilbert, 1926) and "In opposition to the foolish Ignoramus (we will not know; i.e., intuitionists), our slogan shall be: We must know – we will know" (Hilbert, 1930; Reidemeister, 1971; Smith, 2014).

Today, we know infinities are important concepts; thus, the Intuitionists were wrong. However, Kurt Gödel showed that the Formalists were also wrong. Unfortunately, there exist true statements which can never be proven (Gödel, 1931, Gödel's incompleteness theorem).

There are many examples in science and mathematics where there have been strong debates and beliefs due to conflicting paradigms. Science historian Thomas Kuhn defines a scientific paradigm as: "universally recognized scientific achievements that, for a time, provide model problems and solutions to a community of practitioners" (Kuhn, 1996).

Rev.1

Time and time again, when there are conflicting paradigms, it is only "for a time". Eventually, we find neither paradigm is true, or both paradigms are true (under a more nuanced understanding). In retrospect, it is more constructive to develop an understanding as to which paradigms may be right under what conditions, as opposed to an all-or-nothing approach of arguing about a singular right paradigm. Alternatively, we could come up with a new paradigm, a new school of thought, a new direction; which replaces or bridges the old way of thinking.

In this paper, we re-examine the current direction and paradigms of interpretability and invite the reader to consider whether it is time for a new paradigm.

## 1.1 Interpretability and faithfulness

Interpretability is the ability to explain a model in understandable terms to humans (Doshi-Velez & Kim, 2017). Model explanations have particularly become important for AI safety, as machine learning is increasingly being used by the industry and affects the lives of most humans. This and additional motivations are elaborated on in Section 2.

Within interpretability, there currently exist two paradigms, called *post-hoc* and *intrinsic* (Lipton, 2018). Section 3 properly describe their stance. Put briefly, the *intrinsic* paradigm believes that only models designed to be explained can be explained (Rudin, 2019). In contrast, the *post-hoc* paradigms believe this constraint is unnecessary and too restrictive to achieve competitive performance (Madsen et al., 2022b).

The position in this paper is that, while both paradigms have yielded some insights on specific domains, their broader impact has been limited because their underlying beliefs are problematic and we should therefore shift our focus towards new paradigms. Section 4 contains the primary support for this position. To prove that new paradigms can be developed, Section 5 then presents three emerging paradigms for interpretability and discusses how they might overcome past challenges, their beliefs, drawbacks, and future directions. However, Section 5 should not be considered a final list of alternative paradigms.

Rev.1/3

Rev.2

Rev.3

At the core of this discussion is how each paradigm approaches *faithfulness*. A faithful explanation means the explanation accurately reflects the model's logic, and ensuring and validating this often presents a major challenge because the model's logic is inaccessible to humans (Jacovi & Goldberg, 2020). Faithfulness is particularly important, as false but convincing explanations can lead to unsupported confidence in models, increasing the risk of AI.

In addition to faithfulness is *comprehensibility*, another equally important desirable (Doshi-Velez & Kim, 2017), measuring how understandable an explanation is to humans (also known as human-groundedness) (Robnik-Šikonja & Bohanec, 2018; Lipton, 2018). However, this position paper focuses primarily on faithfulness, as the paradigms are rooted in faithfulness as discussed in Section 3.2.1, and the issue of comprehensibility often first materializes when considering a specific explanation, this is discussed more in Section 6. However, comments on comprehensibility are made when appropiate throughout the paper.

Rev.1

For these reasons, emerging paradigms (Section 4) attempt to bring new perspectives regarding how to achieve faithfulness. This creates a new opportunity to do interpretability research centered around ensuring faithfulness. However, it also creates a new risk as we may take faithfulness for granted once again, as has been the case with both the intrinsic (Jacovi & Goldberg, 2020) and post-hoc paradigms (Madsen et al., 2022b). To prevent this, this paper also takes the position that we should be vigilant about faithfulness when it comes to new paradigms to prevent repeating past mistakes.

Rev.1

## 2 Why interpretability is needed

Before discussing the current paradigms and their shortcomings, it's necessary to first consider if interpretability is needed at all. Many ethical motivations for interpretability are also served by bias and fairness metrics, so if the current paradigms of interpretability do not work (as we argue in Section 4), perhaps we should drop the idea of interpretability completely. If the models can be made accurate, unbiased, and fair enough, do we need to explain the models? In this section, we will argue that interpretability is required by examining the limitations of bias and fairness metrics and the scientific motivations for interpretability.

## 2.1 Limitations of bias and fairness metrics

There is no doubt that bias and fairness metrics present a vital role in validating models' behavior. However, a shared limitation is that they always measure known attributes (Barocas et al., 2019). For example, gender-bias metrics use gender attributes. This presents two challenges. Can we procure such attributes (known as protected attributes)? How do we prevent unanticipated biases?

### 2.1.1 Protected attribute procurement

Attributes like gender, race, age, disability, etc., are under U.S. law known as "protected attributes" (Xiang & Raji, 2019), and collecting and using these attributes is heavily regulated in most of the world. Andrus et al. (2021) write, "In many situations, however, information about demographics can be extremely difficult for practitioners to even procure.". Therefore, systematically measuring bias and fairness is not always practical (Andrus et al., 2021).

On the other hand, explanations often don't depend on knowing these protected attributes in advance and can provide a more qualitative analysis. For example, consider a résumé screening model, and an adversarial explanation (Ye et al., 2021) which tells us that removing "Woman" from "Member of Woman's Chess Club" changes the prediction from reject to recommend; then this would indicate a potentially harmful bias (Kodiyan, 2019). Therefore, explanations can serve a similar practical purpose to a fairness or bias metric    Rev.1 without performing systematical correlations.

### 2.1.2 Unknown attribute bias

Although protected attributes are important to consider and are often legally protected, many more relevant attributes are involved in ensuring a fair and unbiased system. Unfortunately, it is impossible to consider every possible bias in advance. As an alternative, interpretability offers a more qualitative and explorative validation.

Continuing the example with résumés and automated hiring recommendations, during investigations by Fuller et al. (2021), the authors found that a hospital only accepted candidates with computer programming experience when they needed workers to enter patient data into a computer. Another example was a clerk position where applicants were rejected if they did not mention floor-buffing (i.e., a cleaning method for floors) (Fuller, 2021).

These examples present cases of systematic unintended bias. However, they do not relate to any protected attributes, and they are so specific they can only be discovered through qualitative explanations and investigations. That said, systematic fairness/bias metrics can quantify the damage once potential biases are identified using interpretability. Afterward, those metrics can be integrated into a quality assessment system to prevent future harm.

## 2.2 Interpretability for scientific discovery and understanding

Interpretability is not only used for ethics and adjacent purposes, where bias and fairness metrics have an important role. Interpretability is also used for scientific discovery and learning about what makes models work.

### 2.2.1 Scientific Discovery

An example of scientific discovery is interpretability in drug discovery (Preuer et al., 2019; Jiménez-Luna et al., 2020; Dara et al., 2022). A common approach is to use feature attribution to identify regions in genomic sequences responsible for a particular behavior, such as producing a protein. While these explanations do not guarantee that such connections exist in reality, they can provide important initial hypotheses for scientists enabling them to make more informed choices about the direction of their research.

### 2.2.2 Model understanding

An emerging field of interpretability is mechanistic interpretability, which identifies parts of a neural network that have a particular responsibility (Cammarata et al., 2020). For example, identifying a collection of neurons responsible for copying content in a generative language model, etc. (Elhage et al., 2021). Such insights may not be directly relevant to downstream tasks, but they help us understand current model limitations and can lead to better model design.

| | Intrinsic paradigm | Post-hoc paradigm |
|---|---|---|
| definition | The model is designed to provide explanations by making the explanation part of the model architecture. | The model is produced without regard for explanation, and the explanations are then created after model training. |
| underlying beliefs | Only models that were designed to be explained can be explained. | Although it may be very challenging, black-box models can be explained. |
| | Intrinsic models can have the same performance as a black-box model. | Black-box models will be more generally applicable than intrinsic models. |

Table 1: Comparison of the definitions and underlying beliefs of the intrinsic and post-hoc paradigms. The beliefs relate to a) requirements for a faithful explanation and b) model capabilities. It should be apparent that these two views are seemingly incompatible.

## 3 The current paradigms of interpretability

This paper uses a common definition of interpretability, "the ability to explain or to present in understandable terms to a human" by Doshi-Velez & Kim (2017). However, even this definition of interpretability is not agreed upon.

Lipton says, "the term interpretability holds no agreed upon meaning, and yet machine learning conferences frequently publish papers which wield the term in a quasi-mathematical way" (Lipton, 2018). In 2017, a UK Government House of Lords review of AI noted after substantial expert evidence that "the terminology used by our witnesses varied widely. Many used the term transparency, while others used interpretability or explainability, sometimes interchangeably" (House of Lords, 2017, 91).

To the credit of the field, there have been many attempts at rectifying this with unified taxonomy Mohseni et al. (2021); Ali et al. (2023); Graziani et al. (2023). Unfortunately, there is still no universally agreed-upon definition of interpretability, nor the current paradigms of interpretability (Carvalho et al., 2019; Flora et al., 2022). As such, this section defines the *intrinsic* and *post-hoc* paradigms, as well as describe their underlying beliefs, which are summarized in Table 1.  Rev.1

### 3.1 Definitions

Jacovi & Goldberg (2020) write: "A distinction is often made between two methods of interpretability: (1) interpreting existing models via post-hoc techniques; and (2) designing inherently interpretable models. (Rudin, 2019)". Based on this and other sources (Schwalbe & Finzel, 2024; Dang et al., 2024; Molnar, 2020; Bonifácio, 2024; Madsen et al., 2022b; Arya et al., 2019; Carvalho et al., 2019; Murdoch et al., 2019), this paper refers to these two ideas respectively as 1) the *post-hoc* paradigm and 2) the *intrinsic* paradigm.  Rev.1

#### 3.1.1 The intrinsic paradigm

The intrinsic paradigm works on creating so-called *inherently interpretable models*. These models are architecturally constrained, such that the explanation emerges from the architecture itself.

Classical examples are decision trees, linear regression, and prototypes (e.g. kNNs, Fix & Hodges 1951; Bien & Tibshirani 2009; Blei et al. 2003). In the field of neural networks, some examples are: 1) Generalized Additive Models (Agarwal et al., 2021; Lou et al., 2013; 2012) 2) Attention-based feature attribution (Bahdanau et al. 2015, Section 5.2.1; Luong et al. 2015, Appendix A; Vaswani et al. 2017, Appendix; Jain & Wallace 2019), where attention points to which input tokens are important. 3) Concept bottlenecks (Koh et al.,

2020; Zarlenga et al., 2022) 4) Neural Modular Networks (Andreas et al., 2016; Gupta et al., 2020; Fashandi, 2023), which produce a prediction via a sequence of sub-models, each with known behavior. 5) Prototypical Networks (Kim et al., 2014; Alvarez-Melis & Jaakkola, 2018; Chen et al., 2019), which predicts by finding   Rev.1 similar training observations.

Occasionally, the term "*ante-hoc*" is used instead of *intrinsic*, where "*ante-hoc*" means anything that isn't post-hoc (Retzlaff et al., 2024). This is much more encompassing than just architecturally constrained models, including also models with changes to their optimization procedure. However, such categorization is unsuitable when discussing paradigm shifts, as it's so encompassing that there are by definition no other paradigms. Additionally, *intrinsic* captures more precisely the current literature; for example, almost all interpretability surveys only discuss the post-hoc paradigm or architecturally constrained models (i.e. the intrinsic paradigm) (Retzlaff et al., 2024; Schwalbe & Finzel, 2024; Dang et al., 2024; Molnar, 2020; Bonifácio, 2024; Madsen et al., 2022b; Arya et al., 2019; Carvalho et al., 2019; Murdoch et al., 2019).   Rev.1

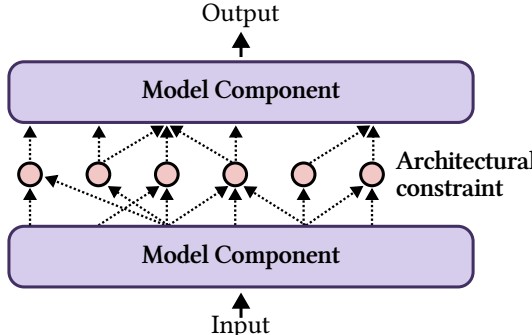

Figure 1: Abstract diagram of the intrinsic paradigm, where the model is architecturally constrained, such that the constraint itself is the explanation. In cases of Decision Trees the entire model is constrained, but often (e.g. Prototype Networks or Attention) only part of the model is constrained.

### 3.1.2   The post-hoc paradigm

*Post-hoc* explanations are computed after the model has been trained. They are developed independently of the model's architecture and how it was trained. However, there are often some simple criteria, like "the model should be differentiable", "the training dataset is known", or "inputs are represented as tokens" (Madsen et al., 2022b). Although general applicability is technically not a requirement, if a method is so specific that it only works on one specific model, it's likely an *intrinsic explanation*.

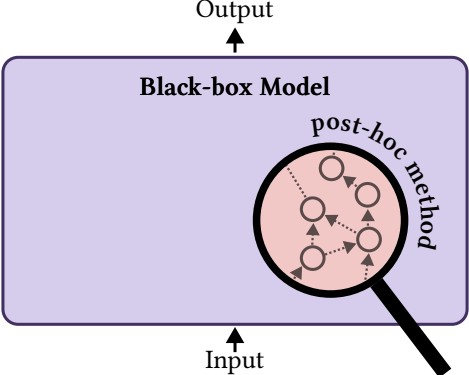

Figure 2: Abstract diagram of the post-hoc paradigm, where a post-hoc method is used to explain a black-box model. The post-hoc method is usually an algorithm, like the gradient w.r.t. the input, but it can also be an auxiliary model.

Common examples in the field of neural networks are: 1) gradient-based or occlusion-based feature attribution, such as Integrated Gradient (Sundararajan et al., 2017), Shapely approximations (Lundberg & Lee, 2017), LIME (Ribeiro et al., 2016), and Grad-CAM (Selvaraju et al., 2020), which indicates what input features are important. 2) Influence functions Koh & Liang (2017), which indicates what training observations are important. 3) Probing methods, such as BERTology (Belinkov & Glass, 2019; Belinkov et al., 2020), which show where information or concepts are stored. 4) Surrogate models (Alizadeh et al., 2020; Kazhdan et al., 2020), such as LORE (Guidotti et al., 2018), DeepRED (Zilke et al., 2016), and BETA (Lakkaraju et al., 2017), which distill the black-box model to an inherently interpretable model like decision tree (Craven & Shavlik, 1995). 5) Concept discovery, such as T-CAV (Kim et al., 2018) and ACE (Ghorbani et al., 2019), which identifies abstract properties which are relevant to the classification, like stribs' relevance in zebra-classification. All of these explanations apply to general models after training. For example, gradient-based feature attribution methods work by differentiating the prediction with respect to the input. The idea is that    Rev.1 if a small change in input causes a big change in the output, then that input is important (Baehrens et al., 2010; Karpathy et al., 2015; Seo et al., 2018).

## 3.2 Beliefs

As with all paradigms, there are fundamental underlying beliefs, which are why the paradigm's followers partake in their paradigm of choice. At the core of these beliefs are two central questions. When are explanations faithful and what are the requirements for faithfulness? And, how do these requirements affect the model's general performance capabilities? This section defines these beliefs and the motivations that lead to them, section Section 4 then discusses the limitations of the intrinsic and post-hoc paradigms in the context of these beliefs.    Rev.1

### 3.2.1 When are explanations faithful?

The intrinsic paradigm believes that: *only models designed to be explained, can be explained,* and the only approach to achieve this is via architectural constraints, i.e. *inherently interpretable models.* Specifically, they argue that using black-box models and post-hoc explanations is dangerous, as these models and methods do not guarantee faithfulness (Rudin, 2019).    Rev.1

The *post-hoc* paradigm takes a less strict stance and believes that even models that were not designed to be explained (i.e., black-box models) can still be explained. Although faithful post-hoc explanations may be much more challenging to produce, *post-hoc* paradigm believes it's possible.    Rev.1

In conclusion, the intrinsic paradigm considers explanations to be part of the model design, and post-hoc explanations are always applied after the model design. Madsen et al. (2022b) frame intrinsic as proactive and post-hoc as retroactive. Hence, the two schools of thought are incompatible frameworks, and they can philosophically be considered as paradigms (Kuhn, 1996).

### 3.2.2 What is the effect on the model's general performance capabilities?

It would seem that *intrinsic explanation* is the obvious choice. If we can control the model such that the faithfulness of explanations can be guaranteed, why consider *post-hoc explanation*?

The commonly mentioned idea is that the *post-hoc* paradigm believes that by constraining the models in the manners that the *intrinsic paradigm* requires, there is a trade-off in performance (DARPA, 2016). However, this trade-off does not have to be the case in practice (Rudin, 2019, section 2).

A more accurate take, which is rarely explicitly discussed, is that the common industry prefers off-the-shelf general-purpose models and only later thinks about interpretability (Bhatt et al., 2019). Additionally, most research only considers predictive performance, not interpretability. Therefore, *intrinsic* researchers are always catching up to black-box models. From the *post-hoc* perspective, it would make more sense to work on generally applicable interpretability methods for both off-the-shelf and future black-box models.

From the intrinsic perspective, while the industry might prefer off-the-shelf models now, they shouldn't. Not validating models through intrinsic explanations can have serious consequences (Rudin, 2019) and eventually

damage their business. Additionally, with increasing legal requirements to provide explanations, the industry may have to use inherently explainable models (Goodman & Flaxman, 2017).

For these reasons, the *intrinsic* paradigm believes we should not let the industry's needs dictate our research direction, as their goals may be too short-sighted. In the long run, intrinsic models may be the only reasonable option.

In conclusion, the *post-hoc* paradigm has good intentions of providing general explanations for general-purpose models. However, from the *intrinsic* paradigm perspective, those good intentions are meaningless if it is fundamentally impossible to provide guaranteed faithful explanations without an *inherently interpretable model*.

## 4 Why interpretability needs a new paradigm

It tends to be the case that when there are multiple paradigms, it is because neither of the paradigms fits the needs. However, for the case of the *post-hoc* and *intrinsic* paradigms, it could be argued that they serve different needs. For example, *intrinsic* explanations should be preferred for critical applications (Rudin, 2019), and *post-hoc* explanations could be used for verifiable situations, such as drug discovery, where the hypothesis generated by the explanations is verified using physical experiments.

### 4.1 The case against the intrinsic paradigm

The industry primarily uses post-hoc explanations, including for high-stakes applications such as insurance risk assessment and financial loan assessment (Bhatt et al., 2019; Krishna et al., 2022). This is because such industries usually do not have the in-house expertise to develop custom high-performing inherently interpretable models for their specific task. They must rely on existing inherently interpretable models, which are not generally competitive, or use more advanced off-the-shelf neural black-box models, like pre-trained language models, which will be competitive. In practice, the industry is thus often not in a position to choose inherently interpretable models.     Rev.1

Another challenge with the intrinsic paradigm is that its models are often not completely interpretable because only a part of the model is architecturally constrained to be interpretable. The rest of the neural network, still use black-box components (e.g. Dense layer, Recurrent layer, etc.) which are not interpretable. As such, the intrinsic promise should not be taken at face value (Jacovi & Goldberg, 2020).

To summarize, intrinsic methods are either not competitive in terms of predictive accuracy, general-purpose enough for the industry (Bhatt et al., 2019), or their intrinsic claims are unsupported (Jacovi & Goldberg, 2020). We will here give a few examples where this can be observed.     Rev.1

An example of a lack of general-purpose performance is General Additive Models (GAMs). GAMs map each input feature via non-linear models to separate latent representations and then combine these via a linear model to the final prediction Lou et al. (2012). GAMs have been used successfully in practice (Caruana et al., 2015; Lou et al., 2013), sometimes by extending it to all feature-pairs (Schug et al., 2023). The limitation is that they only work well on tasks that do not require high-order combinatorial feature modeling, which is unfortunately often the case.     Rev.1

An example of unsupported faithfulness is classic attention-based models. Attention itself is interpretable, as it's a weighted sum, and explains the importance of each intermediate representation. However, attention is often used as token-importance (Bahdanau et al. 2015, Section 5.2.1; Luong et al. 2015, Appendix A; Vaswani et al. 2017, Appendix; Jain & Wallace 2019). This is not faithful, as the intermediate representations are produced by a black-box recurrent neural network (e.g. LSTM Hochreiter & Schmidhuber 1997) which can mix or move the relationship between tokens and the intermediate representations. Therefore, the attention scores do not necessarily represent token-importance (Bastings & Filippova, 2020).

Another example is Neural Modular Networks, which produce an executable problem composed of sub-networks, such as `find-max-num(filter(find()))`, which is interpretable (Fashandi, 2023; Andreas et al., 2016; Gupta et al., 2020). However, each sub-networks (`find-max-num`, `filter`, `find`) is itself a black-box

model with little guarantee that it operates as intended (Amer & Maul, 2019; Subramanian et al., 2020; Lyu et al., 2024).

The dynamic between faithfulness and general-purpose predictive accuracy is often observed with concept bottlenecks (Koh et al., 2020; Zarlenga et al., 2022), where a layer in a neural network restricts the intermediate representation to activations of pre-determined concepts; for example, wing-color, beak-length, etc. in a bird-classification task. This requires all relevant concepts to be known and labeled. However, this is rarely satisfied and works have shown that the concepts leak extraneous information unrelated to the concepts (Margeloiu et al., 2021; Mahinpei et al., 2021). Recent works have attempted to control this leakage by allocating vector space for unknown concepts, but their faithfulness is still lacking (Ismail et al., 2024).    Rev.1

Overall, there are few success stories within the intrinsic paradigm, where intrinsically faithful explanations have been provided without impacting predictive accuracy and model generality.    Rev.1

### 4.2 The case against the post-hoc paradigm

Although post-hoc explanations directly address the interpretability challenge of black-box components and models, and could therefore provide more complete explanations, there are also few success stories with post-hoc, where post-hoc explanations are consistently faithful.

Most notable is perhaps post-hoc feature attribution explanations (also known as importance measures, IMs),  Rev.1 where the explanation indicates which input features are the most important for making a prediction. The pursuit of such explanations has produced countless papers (Erhan et al., 2009; Štrumbelj & Kononenko, 2014; Zeiler & Fergus, 2014; Karpathy et al., 2015; Li et al., 2016; Shrikumar et al., 2017; Smilkov et al., 2017; Ahern et al., 2019; Thorne et al., 2019; ElShawi et al., 2019; Sangroya et al., 2020), among the most popular are methods like LIME (Ribeiro et al., 2016), Shapely approximations (Lundberg & Lee, 2017), Grad-CAM (Selvaraju et al., 2020), and Integrated Gradient (Sundararajan et al., 2017).    Rev.1

However, repeatedly, the faithfulness of these IM explanations is criticized (Adebayo et al., 2018; 2021; Kindermans et al., 2019; Hooker et al., 2019; Yeh et al., 2019). For example, there is great disagreement between alleged faithful IMs, which is hard to reconcile (Jain & Wallace, 2019; Krishna et al., 2022), and they are not robust to adversarial model attacks (Bordt et al., 2022; Slack et al., 2020). Other works show their faithfulness is both task and model-dependent and thus don't provide the generality that the *post-hoc* paradigm desires (Bastings et al., 2022; Madsen et al., 2022a). Finally, theoretical works suggest that IMs are subject to a *no free lunch theorem* (Han et al., 2022), or it may be impossible to provide faithful post-hoc IMs (Bilodeau et al., 2024).

Similar to the work of IM, is the visualization of neurons in computer vision, which shows that neurons represent high-level concepts, such as nose or dog. This is done by visualizing convolutional weights or the input image that maximizes a neuron's activation (Olah et al., 2017; Nguyen et al., 2016; Yosinski et al., 2015), which provides very convincing evidence. However, it has been shown empirically, theoretically, and through human-computer-interaction (HCI) studies that these visualizations do not provide comprehensible explanations regarding the neurons' responsibility (Geirhos et al., 2023; Borowski et al., 2021; Zimmermann et al., 2021)[1].    Rev.1

Another notable example is probing explanations, where models are verified by relating the model's behavior or intermediate representation to, for example, linguistic properties (part-of-speech, etc.) (Belinkov & Glass, 2019; Belinkov et al., 2020). This idea has produced an entire subfield called BERTology (Rogers et al., 2020). BERTology in particular has attained substantial attention (Coenen et al., 2019; Clark et al., 2019; Rogers et al., 2020; Clouatre et al., 2022; McCoy et al., 2019; Conneau et al., 2018; Tenney et al., 2019), with most of the works finding that neural networks can learn linguistic properties indirectly.

Unfortunately, like post-hoc feature attribution, there are many reasons to be highly skeptical (Belinkov, 2021). For example, using an untrained model or a randomized dataset shows an equally high correlation with linguistic properties, compared with training a regular model (Zhang & Bowman, 2018; Hewitt & Liang,

---

[1]Neural networks likely do encode high-level concepts, but these visualizations are not effective (i.e. low comprehensibility) for identifying the responsibility of specific neurons.

| | Learn-to-faithfully-explain paradigm | Faithfulness measurable model paradigm | Self-explaining model paradigm |
|---|---|---|---|
| definition | The model is optimized such that an explanation method becomes faithful. | The model is designed to enable measuring faithfulness of a category of explanations. | The model directly outputs both its prediction and an explanation for that prediction. |
| underlying beliefs | The relaxed faithfulness metric used for optimization is a sufficient approximation. | It is computationally feasible to optimize explanations for optimal faithfulness. | Models can be trained to model and articulate their own reasoning accurately. |
| | Models can be optimized such that explanations become faithful without losing performance. | Models can be optimized to be faithfulness measurable without loss of predictive performance. | Self-explanation capabilities do not negatively impact regular predictions. |

Table 2: Comparison of the definitions and underlying beliefs of the new paradigms. The beliefs relate to a) explanation requirements and b) model capabilities. These new paradigms can be compared with the old paradigms in Table 1.

2019). These discoveries have put the entire methodology into question, although there is work trying to adapt to these new critiques (Voita & Titov, 2020).

### 4.3 Summary

Post-hoc feature attribution and probing explanations are just two cases where post-hoc shows initial promise through countless papers, only to be debunked repeatedly. The trend is oscillating between proposing new explanation methods and debunking them. Of course, it's impossible to prove that there will never be a great post-hoc method. However, the lack of guarantees also makes it impossible to know when a faithful post-hoc method is proposed. Similarly, intrinsic explanations also receive criticism after a while, as has been the case with attention and Neural Modular Networks.

## 5 Are new paradigms possible?

Although both the intrinsic and post-hoc paradigms have significant issues, parts of their underlying beliefs have merit. The intrinsic paradigm believes that *we can't expect models that were not designed to be explained, to be explained*, while post-hoc believes *black-box models tend to be more general purpose while providing high predictive performance*. These beliefs have merit, and it's worth considering how to incorporate their spirit into new paradigms.

It can seem unlikely that such a paradigm can even exist. However, there is already some work that satisfies these desirables. In particular, we have identified 3 alternative paradigms, summarized in Table 2. These emerging directions have unfortunately not received much focus, likely due to favoritism towards existing paradigms (Kuhn, 1996).

All three paradigms work with what would be black-box models. However, their idea is to optimize these models so that they are designed to be explained. How they differ is in their exact formulation of this approach.

It's important to note, that it's only with hindsight we can truly know if a new idea will become the next major paradigm, and it may be a fourth unknown idea that will become the next major paradigm. As such, the main purpose of this section is not to promote the next paradigm but rather to establish that it is possible to develop new interpretability paradigms.

## 5.1 The learn-to-faithfully-explain paradigm

This paradigm is the most direct application of the optimization idea. An existing explanation algorithm (Bhalla et al., 2023) or model is used (Yoon et al., 2019; Chen et al., 2018), and the predictive model is then optimized to maximize both the predictive performance and the faithfulness. In addition to faithfulness, it's possible to also optimize for comprehensibility-proxies such as sparsity (Bhalla et al., 2023; Jethani et al., 2021). Rev.1

Importantly, this approach does not require the architectural constraints that the intrinsic paradigm applies, as the explanation comes from an external explanation method, not the architectural design. The explanation method can be similar or even identical to those from the post-hoc paradigm. However, because the model is optimized to enable these explanations to be faithful, it's not post-hoc, and there are more reasons to think that the explanations is faithful.

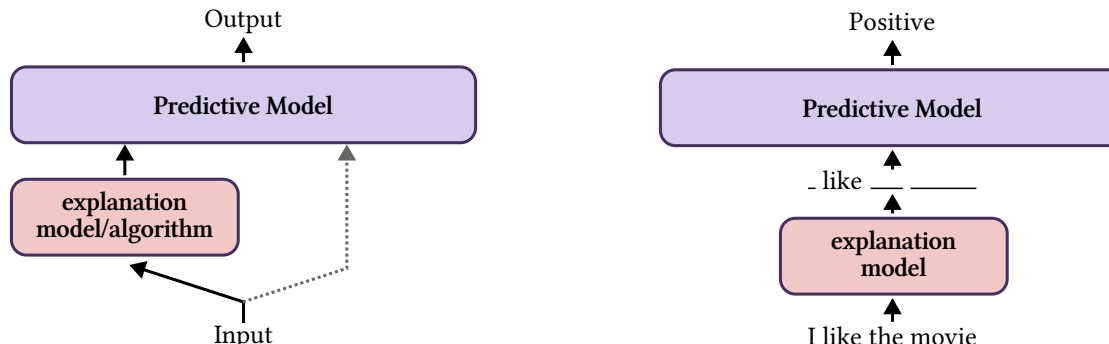

(a) Abstract diagram, showing the joint or disjoined amortized explanation setup. Care needs to be taken that the explanation model doesn't perform the prediction.

(b) Concrete example, where an explanation model produces a feature attribution explanation, showing only "like" is important.

Figure 3: The learn-to-faithfully explain paradigm. In most cases, this paradigm works by generating an explanation from the input, using either a model or an algorithm, this explanation is then fed into the predictive model, which has been optimized to respect the explanation.

Early work on this jointly trains an explanation model and a prediction model (Yoon et al., 2019; Chen et al., 2018). This direction has been called joint amortized explanation methods (JAMs). However, Jethani et al. (2021) point out that the explanation model often learns to encode the prediction, which means the explanation model becomes part of the black-box problem rather than the solution. A solution can be to use a disjoint setup (Jethani et al., 2021), where the explanation model can't encode the prediction, a setup that the following works have adapted (Jethani et al., 2022; Covert et al., 2022). However, the explanation model may still output unfaithful explanations for out-of-distribution inputs. An alternative is to produce the explanation algorithmically (Bhalla et al., 2023), for example by having an explanation algorithm remove unnecessary features, and the prediction model learns to support sparse features.

Regardless of the specific approach used to produce the explanation, the challenges are formalizing the faithfulness objective correctly such that the optimization works as intended, ensuring that the explanations are truly faithful and that the model properties that make explanations faithful also hold for out-of-distribution data (Covert et al., 2022; Bhalla et al., 2023). To validate faithfulness present methods use ground-truths (Jethani et al., 2022; Bhalla et al., 2023) but this is only feasible for simple or synthetic tasks. Another reasonable concern is that adding faithfulness to the optimization objective decreases the predictive accuracy as capacity in the predictive model needs to be used for this objective. As of yet, there is no strong analysis of this concern. However, existing work suggests there are no performance penalties (Covert et al., 2022; Jethani et al., 2022), and the predictive model becomes more robust to adversarial attacks (Bhalla et al., 2023). Rev.1

## 5.2 The faithfulness measurable model paradigm

Normally, measuring faithfulness is extremely challenging (Jacovi & Goldberg, 2020). However, the faithfulness measurable model (FMM) paradigm designs the model such that faithfulness can be easily and precisely measured without requiring architectural constraints. Importantly, because faithfulness is easy to measure Rev.1 by design in FMMs, it's possible to identify the explanation that maximizes faithfulness using optimization algorithms (Zhou & Shah, 2023), which makes the model indirectly intrinsically explainable (Madsen et al., 2024b; Hase et al., 2021; Vafa et al., 2021). In essence, this paradigm reformulates the intrinsic paradigm from 'inherently explainable" to "inherently measurable".

Additionally, because faithfulness can be easily and precisely measured, it's possible to present the faithfulness metric along with the explanation to the user. This can add confidence to the explanation, thus increasing the comprehensibility. While this is technically possible in any paradigm, this paradigm is designed for it.

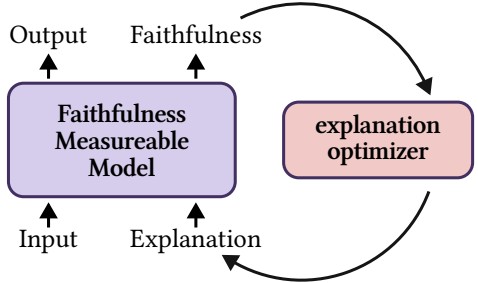 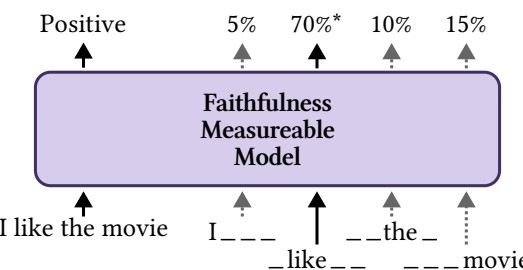

(a) Abstract diagram, showing the optimization loop between the faithfulness measurable model and the optimizer.

(b) Concrete example, showing how the faithfulness of all potential feature attribution explanations is measured, the optimal explanation ("like") is then selected.

Figure 4: The faithfulness measurable model paradigm. In this paradigm, the predictive model can also measure how faithful a given explanation is. The explanation can thus be produced by optimizing an initial (maybe random) explanation towards maximal faithfulness.

Madsen et al. (2024b) and Hase et al. (2021) show that this idea can be achieved using simple data argumentation, and there is no need for architectural constraints. The central idea is to use the erasure metric (Samek et al., 2017) to measure the faithfulness of feature attribution explanations. The erasure Rev.1 metric says: if information (pixels, tokens, etc.) is truly important, then when removing it the prediction should change significantly. The common challenge is that removing information causes out-of-distribution issues (Hooker et al., 2019; Madsen et al., 2022a). However, by using data argumentation during training, it's possible to extend the model to support the partial inputs created by the erasure metric. Importantly this can be achieved without architectural constraints, thus it remains possible to use general-purpose models such as RoBERTa (Madsen et al., 2024b) and GPT-2 (Vafa et al., 2021).

The challenge in this paradigm is about coming up with a way to integrate the faithfulness metric in the model while ensuring there is no performance impact and that the model operates in-distribution (Madsen et al., 2024b). Additionally, developing efficient optimization procedures for optimizing explanations is difficult, due to the discrete nature of many explanations (Hase et al., 2021; Zhou & Shah, 2023).

## 5.3 The self-explaining model paradigm

Rather than using external algorithms or models to produce explanations, Elton (2020) proposes in this paradigm that models should directly output explanations themselves, meaning they become *self-explaining.* This differs from the intrinsic paradigm, as the explanation is produced only by basic inference, not architectural constraints. It is also not post-hoc, as models do not explain themselves without some training towards this. Elton (2020) idea is to consider different sub-models, which output prediction, explanation, and confidence; which all produce these from a latent representation. However, today the most common implementation of Rev.1 this idea is instruction-tuned large language models (e.g., ChatGPT, Gemini, etc.) (OpenAI, 2023; Jiang

et al., 2023; Meta, 2023), which are able to explain themselves in great detail and very convincingly (Chen et al., 2023), because they have been optimized towards this objective (Agarwal et al., 2024).    Rev.1

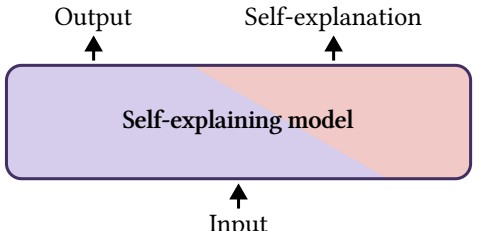 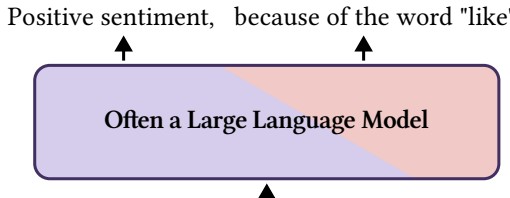

(a) Abstract diagram, where output and explanation is produced by the same model. This idea is not limited to LLMs, Elton (2020) writes about alternatives.

(b) Concrete example, showing a feature attribution explanation saying "like" is important. Feature attributions is not the normal use for LLM self-explanations.

Figure 5: The self-explanation paradigm, where the same model is trained to produce both the regular predictive output and an explanation, called a self-explanation. This paradigm is often seen with Large Language Models, where both the predictive output and the self-explanations appear as generated text.

Unfortunately, because the explanations are produced by a black-box model this paradigm can be quite    Rev.1
dangerous. Therefore, there must be solid evidence that the explanations are faithful for this approach to be valid. However, despite this immediate danger, the model that generates the explanation can in principle have access to all of the logic that produces the prediction. At a minimum, the same weights produce both the prediction and the explanation.

Importantly, self-explanations must relate to the model's reasoning logic, not just the world or abstract concepts. However, presently there is little evidence that this is satisfied (Turpin et al., 2023; Lanham et al., 2023; Madsen et al., 2024a). This is not surprising, as the self-explanations are explicitly trained based on humans' annotating how these explanations should look. However, humans don't have any insight into how the model operates (Jacovi & Goldberg, 2020). As such, the model converges towards very convincing self-explanations with no regard for faithfulness (Agarwal et al., 2024; Chen et al., 2023).

Works addressing the faithfulness challenges of self-explanation are now emerging, with approaches using a combination of in-context learning, self-correction, and post-training to align the model to produce faithful explanations Pan et al. (2024); Chuang et al. (2024); Chen et al. (2024); Paul et al. (2024); Tanneru et al. (2024). However, the improvements have been minor and it's still considered a hard problem (Tanneru et al., 2024). Additionally, just measuring faithfulness of general self-explanation remains a challenge (Huang et al., 2023; Parcalabescu & Frank, 2023; Pan et al., 2024). Despite these challenges, this direction may be worthwhile as self-explanations are a natural solution to generating natural language explanations which are often considered highly comprehensible (Luo et al., 2024; Agarwal et al., 2024).    Rev.1

## 6 Limitations

This position paper primarily focuses on faithfulness, without talking much about comprehensibility. The reason is that the paradigms' underlying beliefs are rooted in concerns regarding faithfulness and performance (Section 3.2) and concerns regarding comprehensibility often first materialize when talking about a specific explanation type, which can generally be produced by any paradigm. For example, feature attributions have been produced within all 5 paradigms ((Bahdanau et al., 2015; Baehrens et al., 2010; Chen et al., 2018; Hase et al., 2021; Huang et al., 2023)) and there is significant HCI literature discussing the comprehensibility of feature attributions (Sen et al., 2020; Schuff et al., 2022b; Rong et al., 2024; Kaur et al., 2020; Gilpin et al., 2018; Prasad et al., 2021; Schuff et al., 2022a; Lertvittayakumjorn & Toni, 2019; Lage et al., 2019).    Rev.1

Additionally, new paradigms on comprehensibility have been proposed. For example, readers are encouraged to study recent works like Schut et al. (2023); Kim (2022), which propose the new idea that it is not enough to frame explanations in terms that humans already understand. We should also develop new language and

mental abstractions for humans to understand machines. Such ideas are orthogonal to this position paper, as they can be applied to any of the paradigms discussed.                                                                    Rev.1

This position paper also does not discuss which specific faithfulness metric to use. This is because which faithfulness metric to use depends on the type of explanation and the paradigms discussed in this position paper cover many or potentially all explanation types. Additionally, developing faithfulness metrics themselves is ongoing research; this is particularly true for the self-explanation paradigm (Turpin et al., 2023; Lanham et al., 2023; Madsen et al., 2024a). Readers are encouraged to study the literature on the meta-evaluation of faithfulness metrics (Hedström et al., 2023; Wang & Wang, 2022), surveys (Zhou et al., 2021), and principles of developing faithfulness metrics (Jacovi & Goldberg, 2020; 2021).                                                   Rev.1

Finally, while this position paper presents five paradigms, it doesn't recommend which paradigm to use. This is because there is presently not enough work to support such a hypothetical claim. Additionally, as discussed in Section 5, it may be that a sixth currently undiscovered paradigm will prevail. Historically, making accurate judgments about paradigm shifts has only been possible in retrospect. It's also possible that some paradigms are better suited for some explanation types, in terms of either faithfulness, comprehensibility, or both. For example, the self-explanation paradigm produces highly convincing natural language explanations (Chen et al., 2023; Agarwal et al., 2024), but is likely ill-suited for feature attribution (Huang et al., 2023). While faithfulness measurable models can provide highly faithful feature attributions, they presently don't optimize for comprehensibility. For these reasons, readers are encouraged to explore such connections and the suitability of each paradigm.                                                                Rev.2

## 7   Conclusion

Although some evidence exists for the emerging paradigms presented in Section 5; these are, first and foremost, just ideas. It's only in retrospect that we can truly know if one paradigm results in meaningful progress in the field. It's also possible the field will converge towards using multiple paradigms, choosing which paradigms to use depending on the application. Alternatively, it's entirely possible that neither of these ideas is what moves the interpretability field forward.                                                            Rev.3 Rev.2

For these reasons, the core position of this paper is that we should shift our attention towards new directions and paradigms in interpretability, be it either entirely new undiscovered paradigms or further developing emerging paradigms; rather than continuing to focus on the existing post-hoc and intrinsic paradigms, which are currently dominating.                                                        Rev.2 Rev.3

That being said, we must also be vigilant regarding faithfulness to avoid repeating past mistakes (Jacovi & Goldberg, 2020). New paradigms present new arguments for why their method is faithful. As we are unfamiliar with these arguments, identifying their flaws is difficult, and it will be easy to get swayed by them.

Historically, a common tactic in post-hoc works was convincing visualization of explanations that aligned with our intuitions (Olah et al., 2017; Yosinski et al., 2015; Nguyen et al., 2016). However, such visualizations are empty arguments, as humans can't know what a true explanation looks like (Geirhos et al., 2023). Likewise, intrinsic works have made seemingly strong theoretical arguments for why their methods are faithful, but these arguments failed to capture the whole model. Even the emerging *learn-to-faithfully-explain paradigm* have already shown sharp corners, where the explainer model unintentionally encodes the prediction and is therefore unfaithful (Jethani et al., 2021).                                                         Rev.3

To prevent false arguments, a sound start is to always have a specific and measurable definition of faithfulness, which works for all methods within a given explanation category (e.g., counterfactual or feature attribution).

Finally, while these emerging paradigms are promising, it's unlikely they will completely erase the current paradigms. For example, we still teach both the particle and wave paradigms in physics; and most scientists don't worry about whether there are true statements in math that cannot be proven.                                Rev.3 Rev.2

Likewise, there will likely always be situations where intrinsic or post-hoc interpretability makes sense. For example, basic statistics and linear regressions can be framed as intrinsic interpretability. Hence, if a company or researchers decide to use a model because of its intrinsically explainable properties, then we should only praise them – as long as they also measure the faithfulness of the explanations.

**Broader Impact Statement**

This paper presents new paradigms for interoperability, focused on ensuring that explanations are faithful to the model. Should this happen, this would have a significant positive impact on society.

However, the paper also discusses existing paradigms, namely post-hoc and intrinsic, citing reasons for why they may not be fruitful directions. This could be problematic if one of those paradigms is truly fruitful, but the field just hasn't found the right method yet, and researchers feel discouraged from working in these directions because of our paper.

Discouraging work is not the intended outcome of this paper. To prevent this, we specifically write in the conclusion that the field should encourage work on interpretability in general, as long as the faithfulness is well supported. As such, there should be no reason for concern regarding this paper's potential negative impact.

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
