# OpenReview forum: "Interpretability Needs a New Paradigm"
_TMLR — Rejected by TMLR_

### Review · Reviewer_9bKt · 2024-12-04

**Summary Of Contributions:**

The authors describe the two main current paradigms in xAI (ante hoc and post hoc), along with the limitations of the current methods belonging to both paradigms from the perspective of faithfulness, without getting into the nature of the explanations themselves.
They then go on to describe three proposed paradigms:
- 1) “learn-to-faithfully explain”. Here, the explanation is first predicted by the model and subsequently used as part of the reasoning. This would generally fall within the ante hoc paradigm.
- 2) “faithfulness measurable model”. Without modifying the model architecture, a model can be made robust to partial masking via data augmentation. Then, the faithfulness of a post hoc xAI method can be verified by removing the input elements deemed inportant. This would be an approach to help verifying post hoc methods.
- 3) “self-explaining model”. This is inherent to generative LLMs, which can be prompted to generate an explanation to any other output. This would behave like a post hoc method where the explanandum and the explainer are the same model.

**Audience:**

Yes

**Broader Impact Concerns:**

Nothing specific.

**Claims And Evidence:**

No

**Requested Changes:**

This manuscript reviews some of the objectives of xAI and presents the advantages and limitations of current ante hoc and post hoc approaches with respect to how they deal with explanation faithfulness, which I appreciated. However, I feel the paper does not deliver in terms of new interpretability paradigms, as set out by the authors. I think this would require to reformulate the paper in order loosen the focus on a paradigm shift.

**Strengths And Weaknesses:**

With an explosion in the number of xAI publications, there is indeed a bit of confusion about what the different objectives of xAI are, and even what an explanation is.
I think that questioning the binary partition between ante and post hoc methods is welcome, since there are many methods that do fall in between.

However, the paradigms proposed in the paper seem more anecdotal than representative of the current trends, and I’m not even sure they actually represent new paradigms. For instance, I have the impression that paradigm 1 is one of the main approaches towards ante hoc interpratability; paradigm 2 represents a training trick to induce better measurability of the faithfulness of post hoc xAI models; and paradigm 3 is more of an artifact of how generative LLMs work rather than an interpretability paradigm (meaning that LLMs can be asked anything and will provide a response, including if they are asked to explain their previous output, which does not mean that this second output should be considered to be an actual explanation at all).

Overall, I commend the authors for their thesis that “we must seek new paradigms for interpretability”, since the loose definitions of what even interpretability or explanability mean may be slowing down the progress towards truly useful interpretability methods. However, I am under the impression that the authors do not fully manage to reach their ambitious objective of proposing such new paradigms, thus making the contribution of the manuscript questionable.

---

> ### Author Response · Authors · 2024-12-29
> **Rebuttal to 9bKt**
>
> Dear Reviewer 9bKt,
>
> Thanks a lot for taking the time to read our paper, it’s clear to us that you have spent significant time on it and also appreciate the claim/position we are trying to convey, namely “we must seek new paradigms for interpretability”.
>
> We also appreciate your skepticism regarding whether any of the three “new” paradigms that we document bring significant value. However, note that we do not claim they bring any value, and only document them to show that developing a new paradigm is in principle possible. In section 5 we write: “It’s only with hindsight we can truly know if a new idea will become the next major paradigm, and it may be a fourth unknown idea that will become the next major paradigm. As such, the main purpose of this section is not to promote the next paradigm but rather to establish that it is possible to develop new interpretability paradigms.”  Therefore, if you think that neither of these paradigms is convincing in their value, that’s a perfectly valid takeaway that strengthens our claim “we must seek new paradigms for interpretability”. Our hope is a reader is then encouraged by our paper to think of a fourth paradigm, which they may deem superior.
>
> Considering this, we only see one divergence between our perception of the paper, namely if the new paradigms we document are “new paradigms”. Please also note, that we do not claim these are new paradigms, we describe them as “emerging” paradigms. However, we understand the sentiment of your concern and believe this stems from an understandable misunderstanding regarding the definitions of the existing paradigms, namely your use of “ante-hoc” and our use of “intrinsic”. We apologize for not mentioning ante-hoc anywhere in the paper. We have corrected this by properly contrasting ante-hoc with our definition of “intrinsic” in section 3.1.1.
>
> We assume ante-hoc is defined as “incorporating interpretability techniques directly into the model architecture or learning process” [1]. By this definition, all three emerging paradigms fall under ante-hoc, as they are all based on incorporating interpretability into the learning process. However, we use and define “intrinsic” as “The model is designed to provide explanations by making the explanation part of the model architecture.” (Table 1). The “intrinsic” definition only considers architectural constraints, not a learning process. We apply this more narrow definition as it reflects the vast majority of literature [1,2,3,4,5] and popularized opinions [6]. For example, Molnar [2] writes “Intrinsic interpretability refers to machine learning models that are considered interpretable due to their simple structure”. Additionally, a recent survey on “post-hoc vs ante-hoc” only discusses models that are architecturally constrained when talking about ante-hoc [1]. Therefore, our intrinsic vs. post-hoc definitions enable us to better understand existing methodologies as paradigms and how the emerging paradigms are different.
>
> With these definitions clarified, your concerns regarding the emerging paradigms should then also be resolved:
>
> 1. Learn-to-faithfully explain: Is indeed ante-hoc via a “learning process”. It’s also the least “new”, as the earliest work is from 2018, but we still see the field being dominated by post-hoc and intrinsic methods. For that reason, we consider it a demonstration of new paradigms being possible to develop.
>
> 2. Faithfulness measurable model: This is not post-hoc but ante-hoc, as it enables cheap and accurate faithfulness metrics via a “learning process”, which turns the explanation generation into an optimization problem. The earliest work is from 2021.
>
> 3. Self-explaining model: We agree that current popular LLMs operate in a pseudo-post-hoc manner, and we also warn against this in section 5.3. However, recent works train the LLMs such that the self-explanations are faithful, which makes them allegedly ante-hoc via a “learning process” [7,8,9,10], although it’s not a solved problem. We have added more citations and discussion of these recent works. We have also added details regarding now this paradigm is not exclusive to LLMs.
>
> To summarize, all three paradigms are ante-hoc via a “learning process”, do not apply architectural constraints, and formulate the explanation process in very different manners (auxiliary explanations, optimizing explanations, and self-explanations), as is also mentioned in the intro of section 5. Therefore, we consider them a sufficient deviation from the currently dominating “intrinsic vs post-hoc” narrative and as separate paradigms.
>
> We really appreciate your support of our position and feedback. We have added a discussion on ante-hoc vs intrinsic, cited additional recent works on self-explanations, and more broadly clarified these are emerging paradigms. We hope these changes address your concerns and you will consider updating your “Claims And Evidence” score to reflect this. Thanks again.

---

> ### Author Response · Authors · 2024-12-29
> **References regarding "Rebuttal to reviewer 9bKt"**
>
> ### References:
>
> [1] Retzlaff, C. O., Angerschmid, A., Saranti, A., Schneeberger, D., Röttger, R., Müller, H., & Holzinger, A. (2024). Post-hoc vs ante-hoc explanations: xAI design guidelines for data scientists. Cognitive Systems Research, 86(June 2023), 101243. https://doi.org/10.1016/j.cogsys.2024.101243
>
> [2] Molnar, C. (2020). Interpretable Machine Learning. A Guide for Making Black Box Models Explainable. https://christophm.github.io/interpretable-ml-book
>
> [3] Schwalbe, G., & Finzel, B. (2024). A comprehensive taxonomy for explainable artificial intelligence: a systematic survey of surveys on methods and concepts. Data Mining and Knowledge Discovery, 38(5), 3043–3101. https://doi.org/10.1007/s10618-022-00867-8
>
> [4] Bonifácio, S. M. M. (2024). Explainable AI: A case study on a Citizen’s Complaint Text Classification Model. Ph.D. thesis at Universidade de Brasília. http://repositorio.unb.br/handle/10482/50962
>
> [5] Dang, Y., Huang, K., Huo, J., Yan, Y., Huang, S., Liu, D., Gao, M., Zhang, J., Qian, C., Wang, K., Liu, Y., Shao, J., Xiong, H., & Hu, X. (2024). Explainable and Interpretable Multimodal Large Language Models: A Comprehensive Survey. 14(8), 1–33. http://arxiv.org/abs/2412.02104
>
> [6] Rudin, C. (2019). Stop explaining black box machine learning models for high stakes decisions and use interpretable models instead. Nature Machine Intelligence, 1(5), 206–215. https://doi.org/10.1038/s42256-019-0048-x
>
> [7] Chuang, Y.-N., Wang, G., Chang, C.-Y., Tang, R., Yang, F., Du, M., Cai, X., & Hu, X. (2024). Large Language Models As Faithful Explainers. http://arxiv.org/abs/2402.04678
>
> [8] Chen, Y., Singh, C., Liu, X., Zuo, S., Yu, B., He, H., & Gao, J. (2024). Towards Consistent Natural-Language Explanations via Explanation-Consistency Finetuning. ArXIv. http://arxiv.org/abs/2401.13986
>
> [9] Paul, D., West, R., Bosselut, A., & Faltings, B. (2024). Making Reasoning Matter: Measuring and Improving Faithfulness of Chain-of-Thought Reasoning. 15012–15032. http://arxiv.org/abs/2402.13950
>
> [10] Tanneru, S. H., Ley, D., Agarwal, C., & Lakkaraju, H. (2024). On the Hardness of Faithful Chain-of-Thought Reasoning in Large Language Models. http://arxiv.org/abs/2406.10625
>
> [11] Elton, D. C. (2020). Self-explaining AI as an Alternative to Interpretable AI. Lecture Notes in Computer Science (Including Subseries Lecture Notes in Artificial Intelligence and Lecture Notes in Bioinformatics), 12177 LNAI, 95–106. https://doi.org/10.1007/978-3-030-52152-3_10

---

### Review · Reviewer_6GyU · 2024-12-14

**Summary Of Contributions:**

The paper expresses a position on the need for a new paradigm for interpretability, stating that it should be centered around measuring model faithfullness. The authors backup this position with a discussion of three possible ways of implementing the new paradigm drawing from existing work.

**Audience:**

No

**Broader Impact Concerns:**

No concerns.

**Claims And Evidence:**

No

**Requested Changes:**

The requested changes want to address the weaknesses stated above by making the paper a survey that also expresses a position. In brief: - improving on the structure - extending the recognition of other works - providing more in-depth discussions and arguments

**Strengths And Weaknesses:**

Strengths:
- I totally respect and understand the position of the authors, and I find it important to raise the discussion among the community
- While one may center interpretability around other needs (human-centric explanations, mechanistic interpretations, decomposability, approximation by surrogates, accountability, etc.), the proposal of faithfulness seems a valid and relevant one in terms of strategically positioning research efforts

Weaknesses: I believe the paper is still in an early stage of development, and I will do my best to offer constructive feedback that can help the authors improve it. While I do not take issue with the position presented—since I find it valid and believe every perspective has merit—I will primarily focus on the paper's structure and the quality of the arguments put forward.
- The paper structure can be improved in efficiency by removing repetitions. The concept in Section 1 (Introduction) was already clear after the first paragraph, but it keeps being re-stated with different examples in the subsequent paragraphs. Not being related to interpretability at all, I find this text redundant. Section 3.2.1 seems a summary of Sections 4.1 and 4.2, repeating the same arguments and even using the same references to back them up, e.g. the limitations of attention-based interpretability as a form of intrinsic interpretability, using the same references as support (Jacovi & Goldberg,2020) and (Bhatt et al., 2019). I believe that there is more work to acknowledge than these two papers, and probably more nuances to bring in the discussion than repeating the same arguments. If it is not the case, then I would suggest the authors to just remove the repetitions and go for a single common section that merges 3.2.1 with 4.1. and 4.2 because the points being stated are essentially the same.
- The paper is essentially a survey of existing work up to Section 4, but it fails to provide a comprehensive overview for three reasons: 1. The authors refer to the concept of faithfulness but they do not acknowledge at all the extensive efforts that have been made in meta-evaluations of interpretability methods [1-3]. These works are already strongly claiming the importance of incorporating objectives such as faithfulness and others when taking into account different explanation methods. 2. The authors briefly discuss the dissonance in interpretability definitions, but they barely acknowledge the efforts that have been made in unifying taxonomies such as [4-6] among others, and they only focus on the point of view of a handful of researchers. 3. As faithfulness is central to this work, I would have expected to see methods such as Explainable Boosting Machines [7-8] and Shapley-GAM [9-10] discussed.
These works [1-3] and [7-10] have already made a point about the importance of faithfulness and have even suggested methods that get close to ideal [7-10]. This makes me think that the shift of paradigm the authors are proposing is not so drastic and could in fact already have been envisioned by many other researchers that the authors have not acknowledged. It’s not uncommon for similar ideas to emerge independently, so this can certainly happen. However, I believe it would be beneficial to extend the review of related work, as acknowledging these prior contributions could strengthen the argument and provide a fuller context for the proposed paradigm shift.
- The argument in 5.1 does not really discuss the trade-off between sacrificing some of the model capacity to learn faithful explanations. As we add more than the downstream task loss in the optimisation function, we are working in multi-optimisation frameworks where some of the parameters will need to be sacrificed for the benefit of the additional objective. Essentially this means that there would be a performance loss.  The authors only briefly touch on how generalisation could help to the case.

Another major concern is the nature of the paper and its fit within TMLR. There is no real call for position papers. This is also the reason why I suggest the authors to strengthen the depth and width of their survey work.

1. Hedström, Anna, et al. "Quantus: An explainable ai toolkit for responsible evaluation of neural network explanations and beyond." Journal of Machine Learning Research 24.34 (2023): 1-11.
2. Hedström, Anna, et al. "The Meta-Evaluation Problem in Explainable AI: Identifying Reliable Estimators with MetaQuantus." Transactions on Machine Learning Research.
3. Bareeva, Dilyara, et al. "Quanda: An Interpretability Toolkit for Training Data Attribution Evaluation and Beyond." arXiv preprint arXiv:2410.07158 (2024).
4. Graziani, Mara, et al. "A global taxonomy of interpretable AI: unifying the terminology for the technical and social sciences." Artificial intelligence review 56.4 (2023): 3473-3504.
5. Ali, Sajid, et al. "Explainable Artificial Intelligence (XAI): What we know and what is left to attain Trustworthy Artificial Intelligence." Information fusion 99 (2023): 101805.
6. Mohseni, Sina, Niloofar Zarei, and Eric D. Ragan. "A multidisciplinary survey and framework for design and evaluation of explainable AI systems." ACM Transactions on Interactive Intelligent Systems (TiiS) 11.3-4 (2021): 1-45.
7. Lou, Y., Caruana, R., and Gehrke, J. Intelligible models for classification and regression. In Proceedings of the 18th ACM SIGKDD international conference on Knowledge discovery and data mining, pp. 150–158, 2012.
8. Schug, Daniel, et al. "Extending Explainable Boosting Machines to Scientific Image Data." arXiv preprint arXiv:2305.16526 (2023).
9. Bordt, Sebastian, and Ulrike von Luxburg. "From shapley values to generalized additive models and back." International Conference on Artificial Intelligence and Statistics. PMLR, 2023.
10. Bordt, Sebastian, et al. "Post-hoc explanations fail to achieve their purpose in adversarial contexts." Proceedings of the 2022 ACM Conference on Fairness, Accountability, and Transparency. 2022.

---

> ### Author Response · Authors · 2024-12-29
> **Rebuttal to reviewer 6GyU**
>
> Dear Reviewer 6GyU,
>
> Thanks for investing so much time in reading our paper and providing extensive feedback, it’s a pleasant experience to work with someone who is this engaged. We took all your concerns into account in the revision, and believe we have accommodated them all. However, the paper is first and foremost a position paper, not a survey. Your valuable feedback is therefore incorporated from that perspective. We understand your concern regarding whether TMLR accepts position papers. We had similar concerns but Editor-in-Chief, Kyunghyun Cho, informed us that TMLR does accept position papers: “As long as there are verifiable claims, they are well supported with evidence and there is a reasonable audience for the paper at TMLR, such a position paper is welcome”. Considering this and the changes we have made in accordance with your feedback, we hope you will adjust your scores to reflect this.
>
> **Summary of changes made:**
>
> > 1a. Only one example of paradigm shifts is required in the introduction
>
> We included two examples, one from mathematics and one from science, as interpretability exists in the intersection of these. However, we understand your concern and have removed the science example.
>
>
> > 1b. Section 3.2.1. repeats content from section 4.
>
> Thanks, this is fair feedback. The goal of section 3.2. is to define the beliefs of the paradigms and why they exist, but without judgment. Section 4 then provides a critical discussion regarding the paradigms and their beliefs. We have now removed all judgment-based content from section 3.2.
>
> > 2a. Authors should mention meta-evaluations [4-6]
>
> Thanks for highlighting this. The scope of the position paper is not to say which faithfulness metric to use, as this depends on the explanation and is ongoing research. Therefore, we now mention this in limitation, and provide references to work on meta-evaluations and surveys.
>
> > 2b. Authors should cite efforts in unifying taxonomies of interpretability [1-3]
>
> Thanks, we have added these references and reworded the related paragraphs. Additionally, many additional references have been added to support our definitions of intrinsic and post-hoc.
>
> > 2c. Authors should mention work on Explainable Boosting Machines [7-8] and Shapley-GAM [9-10], and should acknowledge existing work on paradigm shifts.
>
> Indeed there are already works that attempt to provide paradigm shifts to interpretability. Highlighting these works is one of the objectives of section 5. We have now substantially extended section 5.3 on self-explanations with recent works, if you think there are more works we missed we would be eager to know. As for GAM/EBM [7-8], these are inherently interpretable models (i.e. the intrinsic paradigm) which are great when the task only requires pairwise or independent feature modeling, but by definition fail on tasks requiring complex interactions. Thanks for reminding us of GAMs, we have added these works to section 3.1.1 and 4.1. Reference [9] “show that Shapley-based post-hoc explanations of any function implicitly solve the problem of representing the function as a GAM” [9]. This can be used to either understand a post-hoc method (Shapely) or to distill a black-box model into an inherently interpretable model (GAM). We understand the value of this work but believe it is outside the scope of this position paper. Reference [10] discusses the disadvantages of post-hoc methods regarding adversarial attacks and legal frameworks, there is no mention of new paradigms or GAMs. It’s a well-written and relevant paper, so we have added it to section 4.2.
>
> > 3. Talk about generalization and capacity trade-offs in section 5.1
>
> Thanks for raising this concern. We have added content regarding both capacity trade-offs and generalization, specifically regarding predictive performance and robustness to adversarial attacks.

---

> > ### Comment · Reviewer_6GyU · 2025-01-06
> > **Thanks for the clarifications**
> >
> > Dear Authors,
> >
> > Thanks for the clarifications and for checking about the scope of the paper with the Editor in chief! I will then be happy to re-evaluate some of my comments to fit the requirement: “As long as there are verifiable claims, they are well supported with evidence and there is a reasonable audience for the paper at TMLR". With this respect, I will be happy to change Audience: No to Audience: Yes, since there is space for a position paper within the journal, and I know that the topic is relevant to TMLR.
> >
> > As for my other concerns, I just re-read the improved version of the manuscript. Many parts in the paper have been clarified and I find the arguments and claims clearer now, so I am considering reviewing some of the scores. Moreover, several relevant works are now properly referenced and this has also improved the quality of the manuscript by a notch. I leave some additional comments below (for future note, it would be of great help if the changes to the original submission were highlighted with a different colour).
> >
> > One main point is that, in my view, the scope of the paper is still not fully clear in the introduction. Specifically, the position the authors take on the emerging paradigms is not well-defined. The authors mention that new paradigms are emerging, which is discussed in Section 4, but it is unclear why a further argument for the necessity of new paradigms is needed when the literature already suggests their emergence. It would seem more pertinent for the authors to take a clearer stance on one or more of these emerging paradigms, rather than asserting that new paradigms are required, as this point appears to be already supported by the existing literature. One position, for instance, could be that there is "no free lunch" and that paradigms should be chosen on a per-case basis. In that case, the authors could suggest how to choose. Another position, could be that of ranking the existing methods based on faithfulness (Sec 5.2). If so, then the readers could take this as a practical take from the paper. I guess intrinsic algorithms would rank first (such as the inherently interpretable models first) and then rank all the post-hoc methods by their faithfulness scores.
> > To me at this stage the paper is missing the instruments to navigate among the paradigms. It could also be that this is a point that is still unclear, for which maybe then just a discussion could be added in the Limitations.
> > If the authors could make just a small last effort to give practical advice for the readers that goes in line with the position, that would be great.
> >
> > There is a typo in the spelling of "intrinsic" on page 5.

---

> > > ### Author Response · Authors · 2025-01-08
> > > **Answer regarding additional feedback**
> > >
> > > Dear Reviewer 6GyU
> > >
> > > Thanks again for your thorough review. We are happy to hear that you consider the paper to have an audience and that we have addressed all your previous concerns. We also appreciate the new concerns you bring and believe we have made appropriate edits to alleviate them. We are sorry for not highlighting the changes before, earlier changes are now highlighted as “Rev.1” and the new changes based on your most recent feedback are now highlighted as “Rev.2”.
> > >
> > > We believe one unclear thing was the meaning of “new”. When we say “new paradigms” we do so from the perspective of “post-hoc vs. intrinsic” as this is the perspective most readers will have, as such the 3 emerging paradigms we cover are considered “new”. This is now clarified in the introduction.
> > >
> > > We also understand and appreciate your insight that providing a ranking or clearer stance on which paradigm to focus on would be beneficial to the field. However, we don’t believe we can support such a position, as there isn’t yet enough evidence. Indeed what we advocate for is more work in new or emerging paradigms, such claims like this can be made in the future. This is now clarified in limitations and conclusions.
> > >
> > > Your point about no free lunch is also well appreciated, this was already described a bit in limitations where we hypothesize that some paradigms may be best suited for some types of explanations. For example, self-explanations for natural language explanations or faithfulness measurable models for feature attributions. We expanded on this and now include it in the conclusion too.
> > >
> > > We hope this addresses all your concerns and thank you again for your efforts and insights. It’s much appreciated, and we believe the paper has significantly improved due to your feedback.

---

> > > > ### Comment · Reviewer_6GyU · 2025-01-10
> > > > **Thanks for the additional changes**
> > > >
> > > > Dear authors,
> > > >
> > > > Thanks for the additional changes. I see your point on the fact that there is still not enough evidence to take a position on one paradigm rather than another.
> > > >
> > > > Just to reply to one of your comments, I disagree with changing the definition of "new" in the paper. This may be misleading to the readers. "New" is a commonly used English word and it would be rather unusual to adapt its meaning for a one paper. What I would suggest is adjusting the claims of the paper, clarifying that the proposed paradigms are not, in fact, new (or novel). They are rather existing techniques for which the formalisation into paradigms has (i) not been done before or (ii) not been studied enough. In that, I recognise that the contribution of the paper is to formalise such perspective and shift the focus of the community towards these techniques.

---

> > > > > ### Author Response · Authors · 2025-01-15
> > > > > **Thanks for feedback and minor changes**
> > > > >
> > > > > Dear Reviewer 6GyU,
> > > > >
> > > > > Thanks again for taking the time to review our changes for your continued feedback. We have removed the line “From the perspective of the current dominant paradigms, post-hoc and intrinsic, these emerging paradigms are considered new.” and made additional minor changes (marked Rev.3.) to incorporate your feedback.
> > > > >
> > > > > We hope this addresses your concerns, and thank you once again for your feedback.

---

### Review · Reviewer_arMG · 2024-12-23

**Summary Of Contributions:**

This paper argues that most current approaches in interpretable AI research can be placed within one of two paradigms: “intrinsic” and “post-hoc” explainability methods. The authors then posit that although both of these paradigms have noble intentions and are reactions to real needs in AI/society, they lack true verification of the objectives they promise and have failed to be used in real-world tasks compared to their opaque counterparts. As such, this paper argues that interpretability research is in need of new paradigms. The authors discuss three new potential directions that address some of the issues of the existing paradigms (focusing mostly on ensuring model-and-explanation faithfulness) and introduce early examples of works in each direction that show potential promise.

**Audience:**

Yes

**Broader Impact Concerns:**

I do not believe there are any broader ethical implications from this work that would merit a Broader Impact Statement.

**Claims And Evidence:**

Yes

**Requested Changes:**

Below, I list some potential changes for each section and their importance in securing my recommendation for acceptance. If time is limited, please focus on all **critical** requests, followed by all **major** requests. If I misunderstood something at any point, just let me know, as this is always a possibility (apologies in advance if that’s the case!).

### Section 1 (Introduction)

- **(Minor, Typo)** In the Hilbert quote in the introduction, it should be “Cantor” rather than “Candor”
- **(Major, Proper Citation)** The Hilbert quotation discussed above should be properly attributed to a verifiable source. Currently, it is missing a proper reference.

### Section 2 (Why Interpretability is Needed)

- **(Very Minor, Nitpicking, Writing)** I would suggest removing the second comma in “In this section, we will argue that interpretability is required, by examining the limitations of bias and fairness metrics and the scientific motivations for interpretability.” to improve the readability of that sentence.
- **(Minor, Typo)** “Andrus et al. (2021) writes” should probably be “Andrus et al. (2021) write”
- **(Major, Question, Clarification)** I can see a bit of why the authors argue that interpretability could bring you the possibility for vetting a model’s fairness/biases without the need for protected attributes. However, and playing devil’s advocate here, aren’t we simply kicking the can down the road as for one to understand a model’s fairness/biases from an explanation alone, one needs to understand what are some of the protected attributes and detect when they are being used (either directly or through other potentially correlated features)? I understand that looking at a model’s explanations can entirely avoid collecting protected attributes (which is a huge plus), but, say, in the case of the résumé example in this section, couldn’t we vet the model’s fairness by doing a simple find-and-replace for gendered words to a default standard (e.g., they all become their female equivalents) and seeing the effect of that change? This evaluation seems to be probably equally as hard/costly as looking at individual explanations and seeing if gendered terms are deemed important. More importantly, this approach does not require any interpretability or explanations. If this is a valid evaluation approach, then what does interpretability bring that this does not already give you?
- **(Minor, Nitpicking, Typo)** “resume” should be “résumé” across this section’s running example
- **(Critical, Question, Clarification)** Section 1 says that “neither paradigm has been fruitful because their underlying beliefs are problematic and we should therefore look for new directions.” Yet, there are several instances where, today, interpretability has led to improvements or insights in scientific/knowledge discovery. Section 2.2.1 itself provides some of these examples in drug discovery. Two other examples that come to mind are (a) the mathematical proofs obtained by Davies et al. [1] from insights from feature importance methods and (b) the chess strategies learnt by experts from AlphaZero concept-based explanations [2]. In light of this, how can the authors argue that interpretability has not been fruitful? Isn’t that statement contradicting the evidence provided later in Section 2 and the non-exhaustive list of examples I just provided? If I understood this correctly, I would recommend potentially "watering down" this claim to be accurate with respect to the literature showing that certain benefits have come from research already in explainability.

### Section 3 (The current paradigms of interpretability)

- **(Very minor, nitpicking, citation style)** When including the quote of (Doshi-Velez & Kim, 2017), consider citing it using `citet{...}` rather than `citep{…}`.
- **(Major, Intrinsic Missing Works)** The division of inherently interpretable models into “attention”, “Neural Modular Networks”, and “ Prototypical Networks” is, in my opinion, not quite exhaustive.  It misses all of the concept-based literature, which has both intrinsic (e.g., [3 - 6]) and post-hoc methods (e.g., [7 - 9]) and is a highly active area of research today. Moreover, this taxonomy ignores General Additive Models and their neural counterparts (e.g., [10, 11]), which have been used in practice (e.g., [12, 13]). Finally, I would say this distinction should probably include some key works in Prototypical explanations, such as Self-Explaining Neural Networks (SENNs) [14] or traditional topic models (e.g., Dirichlet Latent Allocation (LDA) [15]).

I am aware this paper’s purpose is not to be a survey, so it is ok not to include all relevant works. However, given that part of its argument depends on determining the fallibility of existing approaches, I would say it is important to at least have a more comprehensive view of the area that is being studied. I would appreciate it if the authors could explore a more exhaustive view of intrinsic approaches and adapt their arguments accordingly to build a case for why all of these directions need revision. If that’s possible, then the case made by this work will be much more impactful.
- **(Major, Post-hoc Missing Works)** As above, there are several key post-hoc methods that should be included. At the very least, I would recommend including a citation for the original Saliency approach [16] and some of its more used variants (e.g., [17, 18]), the post-hoc concept-based approaches described above, model-agnostic feature attribution approaches (e.g., [19- 21]), and model distillation approaches (e.g., [22 - 24]).
- **(Major, Clarification, Nitpicking)** It is worth mentioning that attention mechanisms were **not** designed as an inherently interpretable model (as in that was not their original purpose). Considering that, what’s the argument to say that they were an inherently interpretable approach when their original intent was not interpretability, and their analysis, whose interpretability is a not-agreed-upon subject of debate, usually comes in a post-hoc fashion by practitioners as a side-product of its use?

### Section 4 (Why interpretability needs a new paradigm)

- **(Minor, Reference Style)** The reference for (Coenen et al., 2019) seems a bit weird as the author list is repeated multiple times.

### Section 5 (Are new paradigms possible?)

- **(Minor, Clarity)** The first sentence of section 5.2 is a bit difficult to parse, particularly after the comma. Similarly, the follow-up sentences were a bit difficult to parse as a whole (one says, “measuring faithfulness is often extremely challenging”, and the next one says, “faithfulness is easy to measure by design”). After reading them a few times, I think I get what they are trying to say (that the methods discussed in that section make faithfulness easy to measure), but I would suggest potentially rewriting them with the proper connectors to make the flow better and disambiguate potential confusion.
- **(Major, General comment for impact)** To maximize the potential impact of this section, I would recommend extending the Figures (Figures 3-5) to showcase a specific example of each of the paradigms discussed here. That would really help with getting a better understanding of these general directions for people who are not too familiar with works in these subareas (which will likely be the case as attention has been placed in other research directions, as argued in this work).

### General Questions

- **(Critical, Question)** For future paradigms, is it really useful to consider faithfulness without considering human-groundedness or comprehensibility as claimed in Section 6? If so, couldn’t one say that DNNs are already interpretable as if one writes down its forward pass, one can entirely obtain a perfectly faithful explanation for the provided answer even if that explanation is entirely incomprehensible for any human? I would guess the authors believe that DNNs are not interpretable even if one can generate perfectly faithful but incomprehensible explanations for their predictions. If that’s the case, then isn’t there an implicit need for comprehensibility behind any of the outlined paradigms in this paper?
- **(Critical, Question)** How do interpretability works in Causality and Neuro-Symbolic AI fit within the current and potentially new paradigms discussed in this work? I was a bit surprised that there is no mention to either of these two directions, even if they are very popular and highly active areas of work within XAI and interpretable AI. I would, therefore, suggest that they are at least discussed somewhere in this paper or (better) placed within the main argument to strengthen this work’s potential reach/impact.

## References

**[IMPORTANT NOTE] I do not expect in any way for all of these works to be included as part of this paper’s bibliography. I am adding them here just for the sake of completeness/thoroughness and to help add context to my comments above in case the context is helpful.**

- [1] Davies, Alex, et al. "Advancing mathematics by guiding human intuition with AI." *Nature* 600.7887 (2021): 70-74.
- [2] McGrath, Thomas, et al. "Acquisition of chess knowledge in alphazero." *Proceedings of the National Academy of Sciences* 119.47 (2022): e2206625119.
- [3] Koh, Pang Wei, et al. "Concept bottleneck models." International conference on machine learning. PMLR, 2020.
- [4] Chen, Zhi, Yijie Bei, and Cynthia Rudin. "Concept whitening for interpretable image recognition." *Nature Machine Intelligence* 2.12 (2020): 772-782.
- [5] Espinosa Zarlenga, Mateo, et al. "Concept embedding models." NeurIPS 2022-36th Conference on Neural Information Processing Systems. 2022.
- [6] Oikarinen, Tuomas, et al. "Label-free concept bottleneck models." ICLR (2022)
- [7] Kim, Been, et al. "Interpretability beyond feature attribution: Quantitative testing with concept activation vectors (tcav)." International conference on machine learning. PMLR, 2018.
- [8] Ghorbani, Amirata, et al. "Towards automatic concept-based explanations." *Advances in neural information processing systems* 32 (2019).
- [9] Yeh, Chih-Kuan, et al. "On completeness-aware concept-based explanations in deep neural networks." *Advances in neural information processing systems* 33 (2020): 20554-20565.
- [10] Agarwal, Rishabh, et al. "Neural additive models: Interpretable machine learning with neural nets." *Advances in neural information processing systems* 34 (2021): 4699-4711.
- [11] Chang, Chun-Hao, Rich Caruana, and Anna Goldenberg. "Node-gam: Neural generalized additive model for interpretable deep learning." ICLR (2022).
- [12] Lou, Yin, et al. "Accurate intelligible models with pairwise interactions." *Proceedings of the 19th ACM SIGKDD international conference on Knowledge discovery and data mining*. 2013.
- [13] Rich Caruana, Yin Lou, Johannes Gehrke, Paul Koch, Marc Sturm, and Noemie Elhadad. Intelligible models for healthcare: predicting pneumonia risk and hospital 30-day readmission. In *Proceedings of the 21th ACM SIGKDD international conference on knowledge discovery and data mining*, 1721–1730. 2015.
- [14] Alvarez Melis, David, and Tommi Jaakkola. "Towards robust interpretability with self-explaining neural networks." *Advances in neural information processing systems* 31 (2018).
- [15] Blei, David M., Andrew Y. Ng, and Michael I. Jordan. "Latent dirichlet allocation." *Journal of machine Learning research* 3.Jan (2003): 993-1022.
- [16] GradErhan, Dumitru, et al. "Visualizing higher-layer features of a deep network." University of Montreal 1341.3 (2009): 1.
- [17] Selvaraju, Ramprasaath R., et al. "Grad-cam: Visual explanations from deep networks via gradient-based localization." *Proceedings of the IEEE international conference on computer vision*. 2017.
- [18] Sundararajan, Mukund, Ankur Taly, and Qiqi Yan. "Axiomatic attribution for deep networks." International conference on machine learning. PMLR, 2017.
- [19] Ribeiro, Marco Tulio, Sameer Singh, and Carlos Guestrin. "" Why should i trust you?" Explaining the predictions of any classifier." Proceedings of the 22nd ACM SIGKDD international conference on knowledge discovery and data mining. 2016.
- [20] Štrumbelj, Erik, and Igor Kononenko. "Explaining prediction models and individual predictions with feature contributions." *Knowledge and information systems* 41 (2014): 647-665.
- [21] Lundberg, Scott. "A unified approach to interpreting model predictions." NeurIPS (2017).
- [22] Craven, Mark, and Jude Shavlik. "Extracting tree-structured representations of trained networks." Advances in neural information processing systems 8 (1995).
- [23] Zilke, Jan Ruben, Eneldo Loza Mencía, and Frederik Janssen. "Deepred–rule extraction from deep neural networks." Discovery Science: 19th International Conference, DS 2016, Bari, Italy, October 19–21, 2016, Proceedings 19. Springer International Publishing, 2016.
- [24] Lakkaraju, Himabindu, et al. "Interpretable & explorable approximations of black box models." arXiv preprint arXiv:1707.01154 (2017).

**Strengths And Weaknesses:**

Thank you so much for submitting this work! I really enjoyed reading this paper, particularly when it came to the refreshing way it was written compared to standard research works. Below are what I believe are this paper’s main strengths, followed by what I believe are some of its weaknesses:

### Strengths

1. **[Significance] (Critical)** This paper offers an interesting position that may call attention to recent advances in interpretability and build a case for why existing paradigms may benefit from revision. As such, I believe this work has the potential for impact and may be of interest to the wider community.
2. **[Quality] (Major)** The paper’s arguments are generally well-framed and presented. The figures and tables help with the narrative, and the arguments are generally sound. There are some minor inconsistencies/contradictions across the paper (see below); otherwise, the work seems sound and interesting.
3. **[Originality] (Major)** Although several works discuss the fallacies, problems, and inconsistencies across the XAI and interpretability fields, this is the first paper, to the best of my knowledge, that attempts to propose future directions that diverge from those that are currently dominating the space. Because of this, I do believe this work is novel.
4. **[Clarity] (Major)** The paper is well-written and easy to read. There are a few smaller typos (see below), but they are all minor, so the overall flow is unaffected.

### Weaknesses

In contrast, I believe the following are some of this work’s limitations:

1. **[Soundness] (Critical)** I am not entirely convinced that one may focus on faithfulness alone when dealing with interpretability whilst treating comprehensibility as an orthogonal direction. As this work assumes all future paradigms should focus entirely on faithfulness, I would appreciate it if the authors could further elaborate on why comprehensibility should not always be a first-order consideration for any future paradigm. For a specific question on this matter, please see my requested changes.
2. **[Quality] (Major)** There are several gaps in the previous literature discussed in this work, leaving the discussed taxonomy of the existing paradigms incomplete, in my opinion. Please see below for specific questions/concerns about this.
3. **[Quality/Clarity] (Major)** There are a few inconsistencies and contradictions with some points in this paper’s main argument. Please see below for specific questions.

---

> ### Author Response · Authors · 2024-12-29
> **Rebuttal to reviewer arMG**
>
> Dear Reviewer arMG,
>
> We are truly grateful for your appreciation of our work and the extremely extensive feedback you have provided us. We imagine reading the paper in such detail and taking the time to compile all your feedback required a significant time investment, which we humbly appreciate. It’s a pleasure to deal with a reviewer who is this attentive and engaged in improving our paper. Rest assured, we have spent significant time integrating all your suggestions into the revision which we think is a significant improvement.
>
> As your feedback is extensive, we will just be commenting on the major changes. We have made the grammatical edits, added primary source citations for Hilbert, reworked the paragraph in section 5.2, etc., as you insightfully suggest, but do not comment upon it here for brevity.
>
> > 2.3 Couldn’t we vet the model’s fairness by doing a simple find-and-replace for gendered words. If so, then what does interpretability bring?
>
> You are entirely correct. However, what you describe is an adversarial example explanation, which is interpretability. And it’s precisely the kind of explanation we reference, such as removing “Women’s” from “Member of Woman's Chess Club”. We understand the confusion and have edited the content to make it clear this is an explanation.
>
> > 2.5 water down the introduction claim to be accurate with respect to the literature showing benefits have already come from explainability research.
>
> Thanks, you raise a very valid point. We now write “The position in this paper is that, while both paradigms have yielded some insights on specific domains, their broader impact has been limited because their underlying beliefs are problematic and we should therefore look for new directions.”
>
> > 3.2. The list of inherently interpretable models is not quite exhaustive. Missing works on concept-based literature, General Additive Models, and Prototypical explanations.
>
> Thanks for highlighting this. It’s hard to strike a balance between brevity and completeness, so we appreciate your input. We have added most of these references to section 3.1.1, plus additional references for discussing their potential issues in section 4.1.
>
> > 3.3. Post-hoc methods that should also be included: saliency, feature attribution, concept-based approaches, and model distillation approaches.
>
> Thanks again. We think part of the issue here was that we referenced most works by author and not by method name. For example, Ribeiro et al. (2016) without writing “LIME”. Additionally, we were indeed missing works on concepts and distillation. We have fixed all these now.
>
> > 3.4. It is worth mentioning that attention mechanisms were not designed as an inherently interpretable model.
>
> We would somewhat like to disagree with this, although it depends on the definition of “designed”. The original works all showcase attention’s ability to explain their model w.r.t. to input and output words. Bahdanau et al. (2015) in Section 5.2.1, Luong et al. (2015) in Appendix A, and Vaswani et al. (2017) in Appendix (not numbered). We made the references more precise and complete to address this concern.
>
> > 5.2. To maximize the potential impact of section 5, Figures (Figures 3-5) should also showcase a specific example of each paradigm.
>
> Thanks, this is an interesting suggestion. We have added additional graphics with basic concrete examples to the revision.
>
> > Question 1: Is it useful to consider faithfulness without considering comprehensibility as claimed in Section 6?
>
> Thanks for pointing this out. We agree that comprehensibility is important, we now mention this in the introduction. Additionally, we have added discussion on comprehensibility relation to the paradigms in section 5 and 4.2. In section 6 we only meant to say Been Kim’s idea on a paradigm where we expand human language to understand models is orthogonal, not all of comprehensibility. However, faithfulness is still the primary concern in this paper as the paradigm’s beliefs are rooted in faithfulness (section 3.2.1, “When are explanations faithful?”). Additionally, we think comprehensibility primarily comes into play given an explanation type, not a paradigm. For example, feature attribution explanations have been produced with all 5 paradigms but their comprehensibility relates to their sparsity, we elaborate on this in section 6.
>
> > Question 2: How do interpretability works in Causality and Neuro-Symbolic AI fit in the paradigms discussed?
>
> Thanks, we have made revisions regarding this. Neuro-Symbolic AI is now covered more broadly via Neural Modular Networks, Concept Bottlenecks, Rule-based surrogate models, and LLM’s self-explanations. However, we consider the term too broad to reference specifically. Causality we assume refers to how Causal Mediation Analysis can be used as a faithfulness metric, we now reference one work (Paul et al., 2024) in section 5.3. However, as this relates to faithfulness metrics and not explanation paradigms we don’t reference it much.

---

### Review · Reviewer_zJ95 · 2024-12-24

**Summary Of Contributions:**

This paper introduces three new paradigms for interpretability based on explanation faithfulness (i.e. the ability of explanations to correctly explain the inner workings of the model). The paper starts off by introducing the two prevailing paradigms for interpretability - intrinsic (models whose components are inherently explainable) and post-hoc (methods to explain black box models) - and argues that neither of these directly address the issue of faithfulness. The authors then introduce three new paradigms for interpretability based on recent efforts that prioritize explanation faithfulness. The three new paradigms introduced in the paper are: (1) learn-to-faithfully explain (optimizes the model and the explainer together) (2) faithfully measurable (optimizes the explainer directly to maximize faithfulness), and (3) self-explaining (models that can provide both predictions and explanations).

**Audience:**

Yes

**Claims And Evidence:**

Yes

**Requested Changes:**

Please see weaknesses above.

**Strengths And Weaknesses:**

**Strengths:**

The paper provides an interesting discussion about the emergence of new paradigms that center around explanation faithfulness (also known as fidelity in the literature). Although comprehensibility, consistency, etc. are all important aspects of explainablity, the authors do a good job of highlighting the importance of faithfulness and why it might be a good idea to design paradigms around it. Some of the works discussed to position the authors arguments are definitely interesting and would benefit the wider audience to know of them.


**Weaknesses:**

1. Considering how the three paradigms introduced by the paper are centered around explanation faithfulness, it would be nice to have a section that discusses faithfulness in more detail. Although there appears to be no universally agreed upon definition or metric for faithfulness, it would be helpful to the readers to know about related efforts in the area (i.e. attempts to quantify and concretely define what faithfulness means such as [1]) and how they pertain to the paradigms introduced later in the paper.
2. I am skeptical about how the self-explaining paradigm fits in with the other two. To me, there appears to be a large overlap between the intrinsic paradigm and the self-explaining paradigm with the only difference being that the former can generate explanations to reflect certain interpretable architectural constraints. Further, as noted by the authors themselves, these models can provide convincing yet unfaithful explanations that can easily fool human users. In my mind, all surrogate models (please see [2]) should also fit into the way this category as it is currently defined even though they technically belong to the post-hoc paradigm. If the authors want to keep this paradigm as a part of their taxonomy, they should modify this category to sufficiently distinguish itself from the instrinsic paradigm and be less LLM-focused.
3. Although I understand that a meticulously in-depth review of existing methods is beyond the scope of this paper,  I agree with the other reviewers that as a position paper, the readers would benefit from seeing how well-known explainability methods that focus on faithfulness fit into the taxonomy proposed by the authors. In addition to the suggestions by the other reviewers, it might be helpful to include other well-known efforts in this area such as neuro-symbolic methods such as LORE [3], self-correcting LLMs [4], and visual heatmap-based methods like Grad-CAM [5].


**References:**

[1] Zhou, J., Gandomi, A. H., Chen, F., & Holzinger, A. (2021). Evaluating the quality of machine learning explanations: A survey on methods and metrics. Electronics, 10(5), 593.

[2] Alizadeh, R., Allen, J. K., & Mistree, F. (2020). Managing computational complexity using surrogate models: a critical review. Research in Engineering Design, 31(3), 275-298.

[3] Guidotti, R., Monreale, A., Ruggieri, S., Pedreschi, D., Turini, F., & Giannotti, F. (2018). Local rule-based explanations of black box decision systems. arXiv preprint arXiv:1805.10820.

[4] Pan, L., Saxon, M., Xu, W., Nathani, D., Wang, X., & Wang, W. Y. (2023). Automatically correcting large language models: Surveying the landscape of diverse self-correction strategies. arXiv preprint arXiv:2308.03188.

[5] Selvaraju, R. R., Cogswell, M., Das, A., Vedantam, R., Parikh, D., & Batra, D. (2020). Grad-CAM: visual explanations from deep networks via gradient-based localization. International journal of computer vision, 128, 336-359.

---

> ### Author Response · Authors · 2024-12-29
> **Rebuttal to reviewer zJ95**
>
> Dear Reviewer zJ95
>
> Thanks for supporting our paper. We are pleased to work with someone who has such extensive knowledge of the field and is open to our position. We appreciate the significant time you have spent thinking about our position paper and providing feedback. We find all your feedback to be valid and have made improvements to the paper accordingly, which we think have improved the overall quality. We truly appreciate your time on this matter. Below are our comments regarding the changes we have made.
>
> > 1. It would be helpful to the readers to know about efforts on measuring faithfulness (e.g. [1]), and how they pertain to the paradigms introduced later in the paper.
>
> Thanks for this suggestion. To accommodate this, we have added a paragraph to the limitation section, directing readers to works on meta-evaluation, surveys [1], and principles for measuring faithfulness. Additionally, we now mention the specific techniques to measure faithfulness for learn-to-faithfully-explain and faithfulness measurable models, as faithfulness metrics are extra paramount to these paradigms. We don’t go into details elsewhere, as which faithfulness metric to use depends on the explanation types (e.g. feature attribution, data attributions, concepts), and generally we discuss paradigms not explanation types.
>
> > 2. Authors should modify self-explanations to sufficiently distinguish itself from the intrinsic paradigm and be less LLM-focused.
>
> Thanks, you present a very reasonable concern. We have added content on the difference between self-explanation and other paradigms. Additionally, we added an elaboration on Elton’s (2020) original idea, which doesn’t involve LLMs. Finally, we now reference recent works on improving faithfulness within this paradigm.
>
> > 3. Readers would benefit from seeing how well-known explainability methods that focus on faithfulness fit into the taxonomy proposed by the authors.
>
> Another excellent suggestion. We think the issue here was that we reference popular methods by authors but not by method names. We were also missing some key ideas, like concept bottlenecks, GAMs, and surrogate models. We have fixed these issues. Additionally, we now reference [2-5].

---

### Decision · Action_Editor_zg4n · 2025-01-30

**Recommendation:** Reject

**Comment:**

As stated in *Claims and Evidence* and pointed by some reviewers, it is not clear whether this paper truly presents a new paradigm.
On the other hand, I agree with the importance of the faithfulness-oriented XAI research.
Therefore, I would like to suggest that the authors narrow the scope of their claims and restructure the paper around a topic such as "the importance and future directions of faithfulness-oriented XAI research."  The current paper title and writing, which implies the proposal of new paradigms for the entire interpretability research, could be an overstatement.

**Audience:**

This paper discusses the direction of XAI research and falls within the scope of TMLR.

**Claims And Evidence:**

In this paper, the authors proposed new paradigms for interpretability in machine learning.
They particularly highlighted issues with existing XAI techniques from the faithfulness perspective and suggested a few possible paradigms to address these problems.
Unfortunately, some reviewers have expressed concerns about the novelty of these paradigms.
As pointed out in the paper, there is no doubt about the importance of research on faithfulness-oriented XAI.
However, based on the paper and the authors' discussions, I had to conclude that it is unfortunately unclear whether the proposed paradigms are truly innovative.

**Resubmission Of Major Revision:**

The authors may consider submitting a major revision at a later time.